# QualEval: Qualitative Evaluation for Model Improvement

## Abstract

Quantitative evaluation metrics have played a central role in measuring the progress of natural language systems (NLP) systems like large language models (LLMs) thus far, but they come with their own weaknesses. Given the complex and intricate nature of real-world task, a simple scalar to *quantify* and *compare* models is a gross trivialization of model behavior that ignores its idiosyncrasies. As a result, scalar evaluation metrics like accuracy make the actual model improvement process an arduous one. It currently involves a lot of manual effort which includes analyzing a large number of data points and making hit-or-miss changes to the training data or setup. This process is even more excruciating when this analysis needs to be performed on a cross-product of multiple models and datasets. In this work, we address the shortcomings of quantitative metrics by proposing our method QualEval, which enables automated qualitative evaluation as a vehicle for model improvement. QualEval provides a comprehensive dashboard with fine-grained analysis and human-readable insights to improve the model. We show that utilizing the dashboard generated by QualEval improves performance by up to 12% relatively on a variety of datasets, thus leading to agile model development cycles both on open-source and closed-source models and on a variety of setups like fine-tuning and in-context learning. In essence, QualEval serves as an automated data-scientist-in-a-box. Given the focus on critiquing and improving current evaluation metrics, our method serves as a refreshingly new technique towards both model evaluation and improvement.

## 1 Introduction

The recent success of large language models (LLMs) while can be attributed to data and compute scaling, has also been the result of evaluation metrics that allow benchmarking and comparison of models. However, multiple experts have pointed out that these metrics are not sufficient to understand the behavior of LLMs and that they are not a good proxy for real-world performance Liu & Liu (2008); Novikova et al. (2017). While this has created a wave of research work that proposes evaluation metrics, the epidemic of poor evaluation has worsened. Scalar metrics cannot capture the nuances of model behavior, thus making model improvement a time-consuming process. This forces model developers to rely on an army of data scientists and engineers to iterate on models, especially in real-world settings.

In this work, we use "quality over quantity" as a guiding principle to propose our method QualEval, which uses qualitative evaluation to address the issues with quantitative metrics. Given a model that is being developed for a task, QualEval serves as an automated data scientist by analyzing the dataset and the model's predictions to create a comprehensive dashboard containing fine-grained analysis of the model's behavior (Figure 1). The dashboard is accompanied by natural language actionable insights that can be applied to improve the model. QualEval thus significantly saves the time and effort of the model developers while serving as a comprehensive and faithful evaluator.

QualEval's algorithm can be broken down into three steps (Figure 2): (1) *Attribute discovery*: Automatic discovery of domains and sub-tasks in the dataset. (2) *Attribute assignment*: Utilize a novel flexible linear programming solver to assign attributes to instances in the dataset and analyze the performance of the model on different attributes to create a human-readable dashboard. (3) *Insight generation*: Parse the generated dashboard to provide natural language insights that improve

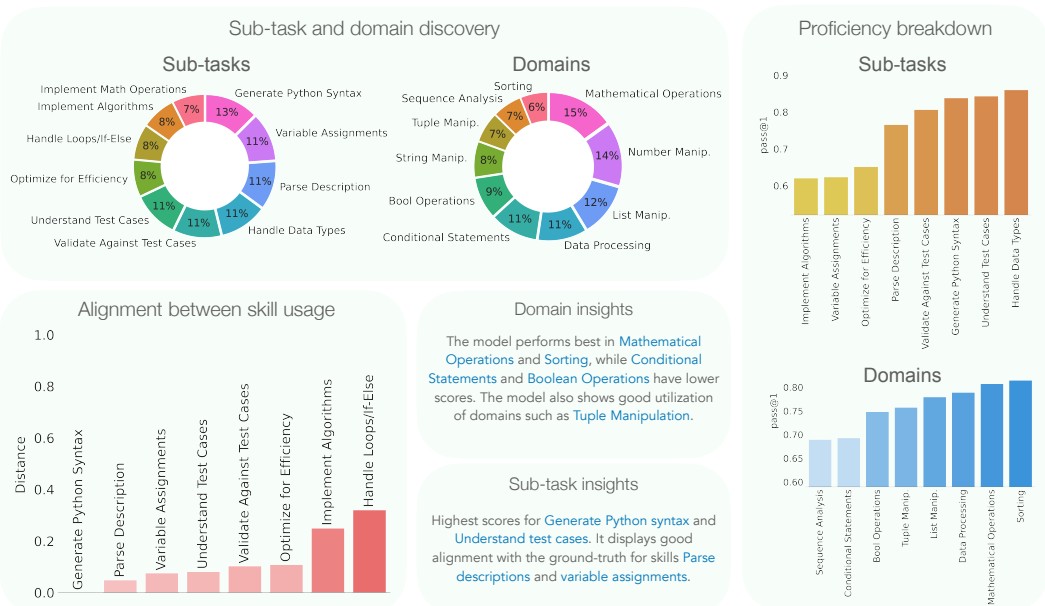

Figure 1: QUALEVAL goes beyond a single scalar metric and provides a dashboard that helps understand the model's performance in a fine-grained manner. The dashboard visualizes the performance of the davinci-3 model on MBPP.

the model. QUALEVAL's end-to-end pipeline is completely automated and requires no human intervention.

QUALEVAL is task-agnostic and we demonstrate its potency on a wide range of tasks including code generation, dialogue summarization, and multiple-choice question answering. QUALEVAL's novel algorithm automatically discovers highly relevant domains and sub-tasks, such as "mathematical operations" and "sorting" for code generation and "Identify the participants" and "Recognize the main topic" for dialogue summarization. QUALEVAL's flexible linear programming solver generates highly accurate assignments, with an 84% and 90% accuracy for domains and sub-tasks assignments when judged by humans. Taken together, QUALEVAL allows the model developer to identify the sub-tasks and domains they need to target while improving the model in the next iteration. For example, on the MBPP dataset for code generation, while the model performs well overall, it performs poorly on the sub-task of "Implement Algorithms", which can be quickly caught and fixed.

QUALEVAL's dashboard is highly interpretable and actionable. We demonstrate that insights from QUALEVAL can be used to *precisely* improve the performance of the open-source Llama 2 model on the selected domains of a dialog summarization task. For instance, the ROUGE-2 score of the Llama 2 model on the "Employment/Professional skills" and the "Career job interview" domains increases by 12 percentage points relatively after applying the insights from QUALEVAL. This allows practitioners to precisely target the domains and sub-tasks they want to improve and therefore could enable unprecedented agility in the model development lifecycle. We additionally improve the ROUGE-2 score of the closed-source curie model from OpenAI by a 10 percentage point relatively by utilizing QUALEVAL's insights to improve the quality of in-context samples and therefore illustrate how practitioners can quickly improve black-box models with QUALEVAL. Therefore, insights from QUALEVAL are *faithful* and *actionable* and can be effortlessly exploited for model improvement.

Our contributions are as follows: (1) We propose the first qualitative evaluation framework for LLMs. (2) We introduce a novel and faithful flexible linear programming-based algorithm to automatically and accurately assign attributes to input instances. (3) We demonstrate that the generated insights are actionable and useful for agile model improvement.

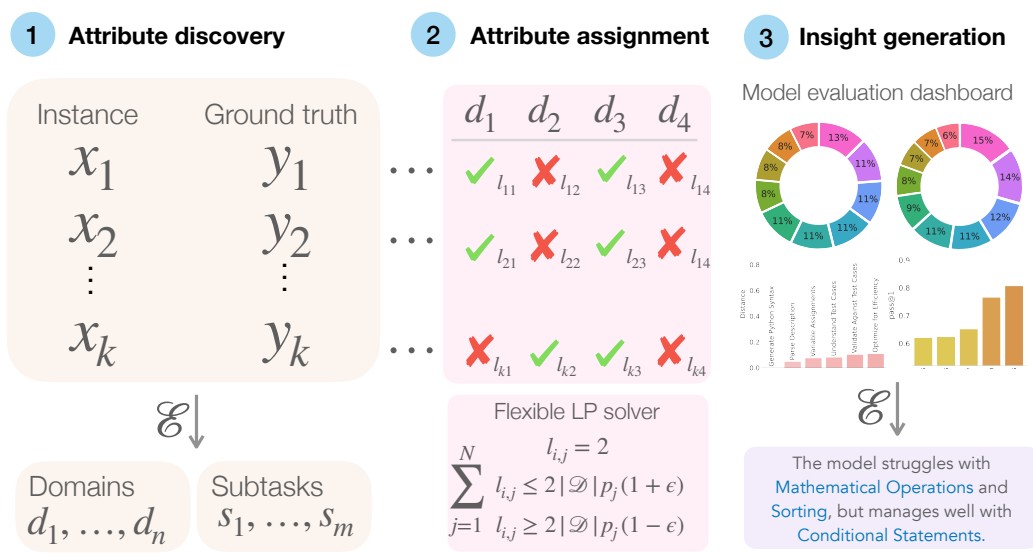

Figure 2: QUALEVAL automatically discovers domains and sub-tasks from input data through an evaluator LLM, $\mathcal{E}$. QUALEVAL then automatically assigns 2 domains and 2 sub-tasks to every sample in the dataset by solving a flexible linear program. Finally, QUALEVAL generates a comprehensive dashboard and presents interpretable and actionable insights for practitioners.

## 2 METHODOLOGY

### 2.1 FORMULATION

**Quantitative evaluation** Quantitative evaluation, which is the standard approach to evaluating models, is typically based on averaging the value obtained by using a metric to evaluate instances of the dataset independently. Formally, given a dataset $\mathcal{D}$ comprising of instances containing inputs ($x_i$) and ground truth outputs ($y_i$), a proficiency metric $\mathcal{M}$, and a model $f$, then:

$$\mathcal{D} \coloneqq \{(x_i, y_i)\}_{i=1}^N$$
$$\hat{y}_i \coloneqq f(x_i)$$
$$\mathcal{M} \colon (x_i, y_i, \hat{y}_i) \to \mathbb{R}$$
$$\underline{\text{Quantitative}} \text{ evaluation} = \frac{1}{|\mathcal{D}|} \sum_{i=1}^{|\mathcal{D}|} \mathcal{M}(x_i, y_i, \hat{y}_i)$$

**Qualitative evaluation** Qualitative evaluation is based on holistically evaluating the model's performance in a fine-grained manner rather than relying on a single scalar value. This could include plots with breakdown of scores for different domains, unique qualitative samples, and human-readable explanations that improve the model. Thus, QUALEVAL outputs a detailed dashboard that describes the intricate nature of the model's performance. Formally, let $\mathcal{V}$ be the vocabulary of the language of the dashboard (here, English) and $\mathcal{I}$ be the set of all possible visualizations. Let $\mathcal{E}$ be an evaluator LLM which generates the dashboard. Then, the output of QUALEVAL is given by:

$$\mathcal{E} \colon \{x_i, y_i, \hat{y}_i\}_{i=1}^N \to (\mathcal{V} \cup \mathcal{I})^\star$$
$$\underline{\text{Qualitative}} \text{ evaluation} = \mathcal{E}\left(\{x_i, y_i, \hat{y}_i\}_{i=1}^N\right)$$

### 2.2 QUALEVAL: QUALITATIVE EVALUATION

QUALEVAL consists of multiple steps that help provide interpretable and actionable insights and we break them down below.

**Attribute discovery** Given the dataset $\mathcal{D}$, QUALEVAL uses an evaluator LLM ($\mathcal{E}$) to automatically discover relevant domains and sub-tasks, $d_1 \cdots d_N$ and $t_1 \cdots t_N$ in the dataset. We refer to these domains and sub-tasks as attributes. Specifically, we prompt $\mathcal{E}$ with the dataset and a task instruction signifying how to solve the dataset ($Instr_D$) to generate the attributes (see A.3 for the exact prompt). Given that datasets can have a large number of instances and LLMs have context length limits, we iteratively sample $k$ instances from the dataset and repeat the prompting process $\frac{|\mathcal{D}|}{k}$ times to generate a large list of attributes ($d_1 \cdots d_M, t_1 \cdots t_L$). To ensure that we choose high-quality attributes, we prune the list of candidates in an iterative process by reducing the size by a factor of $p > 1$ in each turn and repeating the process until we have $N$ attributes. In each step, we prompt $\mathcal{E}$ to shrink the list by choosing the best attributes from the previous list of candidates. Therefore, this iterative scalable procedure allows QUALEVAL to discover attributes in arbitrarily large data across a wide range of tasks, notwithstanding the context window limitations of $\mathcal{E}$.

**Attribute assignment** QUALEVAL performs attribute assignment ($d_1 \cdots d_N$ and $t_1 \cdots t_N$) by scoring the "affinity" or relevance of each instance with different attributes. Let $s_{i,j}^{domain}$ and $s_{i,j}^{task}$ denote the domain and sub-task affinity scores, where $i \in \{1 \cdots |\mathcal{D}|\}$ and $j$ denotes the number of attributes ($\{1 \cdots N\}$).

We use a novel flexible linear programming solver to perform the attribute assignment by ensuring the following properties: (1) An instance is assigned 2 domains and sub-tasks each so that we can give concrete insights. (2) The number of assignments to an attribute is proportional to the prior probability of the attribute. This ensures that rare attributes are not ignored. (3) Choose the assignments with maximum affinity for each instance. We achieve the above wish-list by formulating the attribute assignment as a linear programming (LP) problem.

Given the affinity scores and the prior probabilities, $p_i$, we assign every sample to 2 domains and 2 sub-tasks. However, we want the assignments to respect the prior probabilities i.e. ratio of the number of assignments to all the attributes should be equal to the ratio between the prior probabilities. We enforce this by constraining the number of assignments to an attribute to be $p_i \times |\mathcal{D}| \times 2$.

Let $\mathbf{l}$ be the assignment matrix, where $l_{i,j} = 1$ indicates that the $i^{th}$ sample is assigned to the $j^{th}$ attribute and $l_{i,j} = 0$ indicates otherwise. Let $p_j$ be the prior probability of the $j^{th}$ attribute. To accommodate for the noisiness in an automated method, we make the prior probability constraint flexible by adding some slack, $\epsilon \times p_j \times |\mathcal{D}| \times 2$ ($\epsilon = 0.1$), so that QUALEVAL has some wriggle room to change the attribute probability distribution in favor of better assignments. Therefore, to enforce the prior probability constraint, we sum across the columns of $\mathbf{l}$ and constrain the sum to be between $2 \times |\mathcal{D}| \times p_j \times (1 - \epsilon)$ and $2 \times |\mathcal{D}| \times p_j \times (1 + \epsilon)$. To ensure we assign each sample to 2 attributes, we sum across the rows of $\mathbf{l}$ and constrain the sum to be 2. We formalize the LP as:

$$\max_{\mathbf{l}} \sum_{i=1}^{N} \sum_{j=1}^{N} l_{i,j} s_{i,j}^{domain/task}$$

$$\sum_{j=1}^{N} l_{i,j} = 2 \quad \forall i \in \{1 \cdots |\mathcal{D}|\}$$

$$\sum_{i=1}^{N} l_{i,j} \leq 2 * |\mathcal{D}| * p_j * (1 + \epsilon) \quad \forall j \in \{1 \cdots N\}$$

$$\sum_{i=1}^{N} l_{i,j} \geq 2 * |\mathcal{D}| * p_j * (1 - \epsilon) \quad \forall j \in \{1 \cdots N\}$$

$$l_{i,j} \in \{0, 1\} \quad \epsilon < 1 \quad \forall i, j \in \{1 \cdots N\}$$

We perform an expert verification of the attribute assignments by sampling 100 samples from the dataset and asking three machine learning practitioners if both the domain and sub-task assignments are correct and find that they are indeed correct on average 84% and 90% of the time.

Once we have the assignments, we evaluate each instance using the proficiency metric $\mathcal{M}$ for each domain and sub-task to get $\mathcal{M}(x_i, y_i, \hat{y_i})$. We use the assignments to breakdown the proficiency metric by domains and sub-tasks and automatically generate visualizations that help understand the model's fine-grained performance.

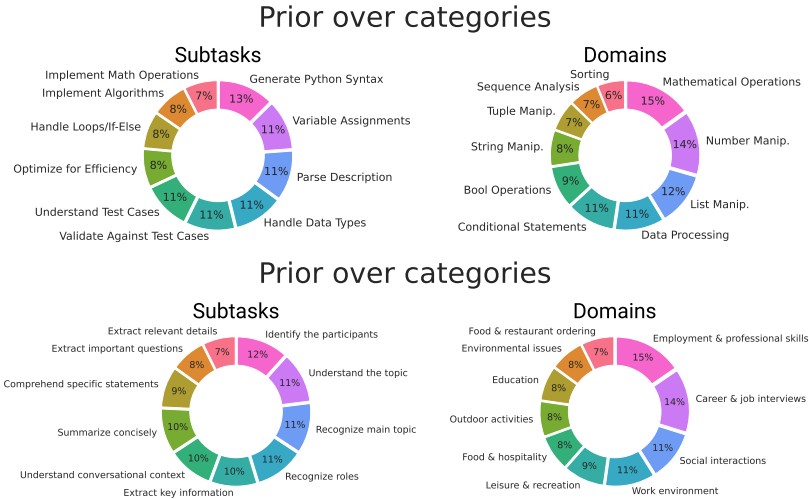

Figure 3: Prior probabilities of domains and sub-tasks on the MBPP and DialogSum datasets

**Measuring sub-task skill alignment** For several datasets, predicting the right answer is not good enough, and producing an answer that uses the same sub-tasks as the ground truth is important. We call this skill alignment and compute it by measuring the correlation between the sub-task affinity scores of the ground truth and the model prediction (higher values implying higher skill alignment).

**Insight generation** QUALEVAL then leverages the visualizations from previous stages to generate useful and actionable insights as a natural language output. We prompt $\mathcal{E}$ with the data from the prior probability, proficiency breakdown, and skill alignment visualizations to generate useful insights (See A.3 for exact prompt). We integrate all the visualizations and insights into a human-readable dashboard depicted in Figure 1.

## 3 EXPERIMENTAL SETUP

**Datasets** We evalaute QUALEVAL on three datasets: MBPP (Austin et al., 2021) (sanitized), DialogSum (Chen et al., 2021), and MMLU (Hendrycks et al., 2020) (clinical knowledge split). We use the same evaluation splits as the original papers and use the test splits for MBPP and MMLU and use the validation split for DialogSum.

**Models** We use both closed and open-sourced models: curie, davinci-2, and davinci-3 models from OpenAI and Llama 2 (13 billion chat models (Touvron et al., 2023)). We use a temperature of 0.9 for all models and use 2 randomly sampled in-context samples when prompting curie, davinci-2, and davinci-3. We instantiate $\mathcal{E}$ with the gpt-3.5-turbo model (OpenAI, 2023).

**Evaluation Metrics** We use the pass@1, ROUGE-2, and accuracy as proficiency metrics for MBPP, DialogSum, and MMLU respectively.

## 4 RESULTS

We systematically present the different visualizations in our reportcard. We first present the attributes (domains and sub-tasks) discovered by QUALEVAL, and visualize their prior probabilities. To holistically understand the model's performance and identify areas of improvement, we leverage the attribute assignments generated by our flexible LP solver, $l_i$, to visualize the average proficiency score of the samples assigned to an attribute. Finally, we present a concise natural language output listing interpretable and actionable insights from the model's performance.

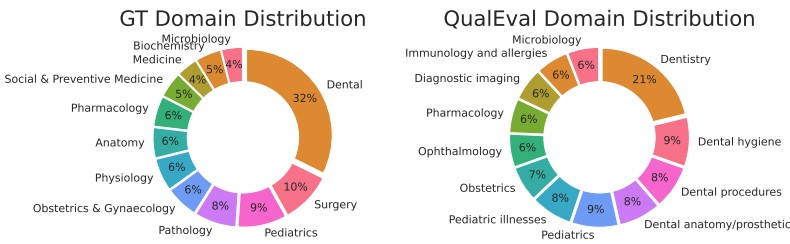

Figure 4: QUALEVAL faithfully discovers and scores attributes. We compare the domain priors discovered by QUALEVAL(right) with the ground truth domain annotations (left) in the MedMCQA dataset and find a high degree of alignment.

### 4.1 PRIOR OVER DOMAINS AND SUB-TASKS

QUALEVAL automatically discovers relevant domains and sub-tasks in the evaluation data and automatically discerns their prior probabilities. Figure 3 presents the prior probabilities of the domains and sub-tasks in the MBPP and DialogSum datasets We find that the MBPP dataset comprises of a large set of samples that involve domains like mathematical/numerical operations (29%) and list manipulation (12%) while domains like sorting (6%) and tuple manipulation (7%) are less prevalent. Interestingly, QUALEVAL captures fine-grained nuances by including closely related yet different sub-tasks like "Implement mathematical operations" and "Implement algorithmic operations", giving practitioners a more nuanced understanding of their evaluation data.

Further, we note that the DialogSum dataset is dominated by samples involving domains like employment and professional skills (15%) and career and job interviews (14%), while domains like education and outdoor activities are less prevalent (8% and 8% respectively). Though the hospitality domain is also frequent, it is listed under two fine-grained domains, "Food and restaurant ordering" (7%) and "Food and hospitality" (8%), which further highlights QUALEVAL's ability to capture fine-grained nuances. The evaluation also suggests the dominance of sub-tasks that involve identifying the participants (12%), understanding and recognizing the main topic (22%), and recognizing the roles in the conversation (11%), which are conceptually important sub-tasks for accurately summarizing a daily conversation between two people.

**Faithfulness of priors**    While most datasets do not have ground truth annotations for the domains and sub-tasks, Pal et al. (2022) introduces a multiple-choice question answering dataset, MedM-CQA, collected from real-world medical exam questions, and importantly includes domain annotations. We randomly sample 250 questions from the MedMCQA dataset and leverage QUALE-VAL to discover domains and find the prior probabilities. We compare the prior probabilities from QUALEVAL with the ground truth domain annotations from MedMCQA in Figure 4. We find that the domain priors from QUALEVAL are highly aligned with the ground truth annotations ("Pediatrics" (9% vs 9%), "Obstetrics and Gynecology"(6% vs 7%), and "Pharmacology"(6% vs 6%) and "Microbiology"(4% vs 6%)). Interestingly, QUALEVAL splits the "Dental" domain into more precise domains such "Dentistry", "Dental Hygiene", "Dental procedures", and "Dental anatomy", further highlighting QUALEVAL's ability to capture hierarchies and nuances in the data.

### 4.2 PROFICIENCY CATEGORIZED BY DOMAINS AND SUB-TASKS

Figure 5 highlights the proficiency of the davinci-3 model on domains like sorting, mathematical operations, and data processing and on sub-tasks like handling data types, understanding test cases, and generating Python syntax. Analysis from Austin et al. (2021) also suggests that models on MBPP perform well on "coding interview" type questions which generally involve data structures, sorting, list manipulation, and data processing, which aligns with QUALEVAL's output.

 Austin et al. (2021) also suggests that models struggle with samples related to advanced math problems and samples with multiple sub-problems. This conforms with QUALEVAL's proficiency breakdown which reveals that the model struggles with samples involving the "Implement algorithms" and

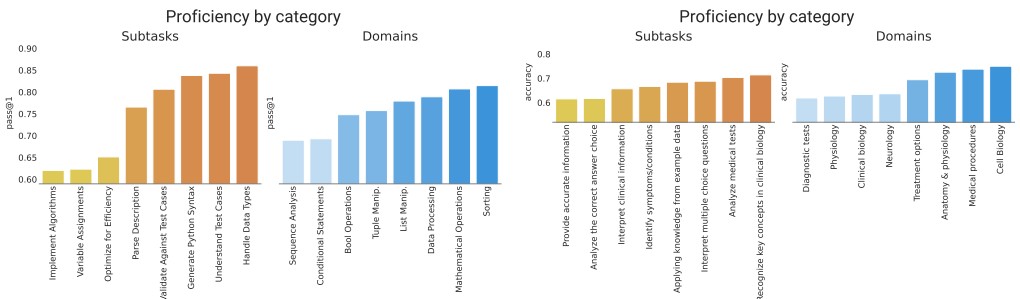

Figure 5: Proficiency breakdown for different sub-tasks and domains in the MBPP and MMLU (clinical knowledge) datasets.

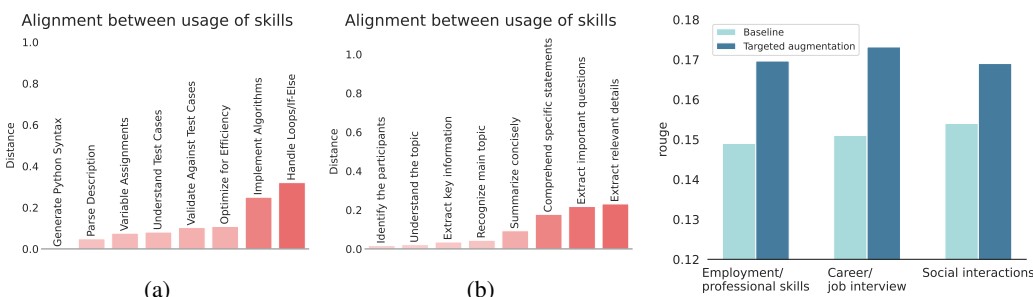

Figure 6: (Left) Model calibration analysis of the davinci-3 model on MBPP and DialogSum. (Right) QUALEVAL enables precise model improvement on the selected domains. We leverage insights from QUALEVAL to improve the ROUGE-2 of the three domains by $\approx 2$ percentage points.

"Variable assignments" sub-tasks and the "Conditional statements" and "Sequence Analysis" domains, which are often leveraged to solve math problems and samples with multiple sub-problems.

QUALEVAL is task-agnostic and potent in niche domains such as clinical data. Figure 5 demonstrates high proficiency of the davinci-3 model on the cell biology and medical procedures domains and sub-tasks related to analyzing and processing medical test data and recognizing key terms/concepts in clinical biology. However, the model struggles with sub-tasks related to providing accurate information and analyzing the correct answer choice.

## 5 ANALYSIS

### 5.1 SKILL ALIGNMENT BETWEEN MODEL GENERATIONS AND GROUND TRUTH

While proficiency metrics like pass@k, BLEU, and ROUGE are able to judge the proficiency of a model, they do not provide insights about model calibration, i.e., whether the model is leveraging the expected subtasks when generating responses. Model calibration is a unique lens to understand model performance, as practitioners could understand how the model generates accurate responses and understand whether the model is generating novel or unexpected solutions.

We therefore measure the calibration of a model by measuring the distance between the affinity scores of model generations and ground truth responses across different subtasks for samples above a certain proficiency threshold (set to $1$ and $0.14$ for the MBPP and DialogSum datasets respectively). We measure the distance between the affinity scores by measuring the fraction of samples where the difference between the affinity scores of the generation and the ground truth is greater than 1. A low distance implies a high alignment between the skills utilized in the model generation and the ground truth response.

Figure 6 highlights the correlation between model generations and ground truth responses for the davinci-3 model on the MBPP and DialogSum datasets. The model on average is highly calibrated

across most attributes on both datasets barring the subtasks about implementing algorithms and handling loops and conditionals. This therefore entails that either the model uses or excludes loops and conditionals when the ground truth response excludes or uses loops and conditionals respectively. We find similar behavior on the DialogSum dataset, where the model is highly calibrated across most attributes, barring subtasks related to extracting relevant details and important questions.

## 5.2 QUALITATIVE SAMPLES

QUALEVAL automatically yields revealing and interesting qualitative samples. To generate revealing qualitative samples, we identify samples where the affinity scores of the ground truth response and model generation are not aligned. This results in an automatic way to generate samples where the model is not calibrated.

Figure 7 shows qualitative samples from the MBPP dataset generated by the davinci-3 and davinci-2 models. The first two examples are generated from davinci-3. In the first example (left), the ground truth program uses XOR to test for uniqueness, while the generation uses a loop to check for uniqueness. In the second example (center), the ground truth program uses an in-built Python function to check equality whereas the model loops through the input to check the condition. Therefore, these examples further validate the finding in the prior section which suggests that the model is not calibrated for handling loops and conditionals.

| | | | |
|---|---|---|---|
| Prompt | Write a python function to find the element that appears only once in a sorted array. | Write a function that takes in a list and element and checks whether all items in the list are equal to the given element. | Write a python function to booleans in the given list |
| Ground Truth | def search(arr):
  n = len(arr)
  XOR = 0
  for i in range(n):
    XOR = XOR ^ arr[i]
  return (XOR) | def check_element(list, element):
  check_element =
    all(v== element for v in list)
  return check_element | def count(lst):
  return sum(lst) |
| Output | def search(arr):
  n = len(arr)
  for i in range(0, n - 1, 2):
    if arr[i] != arr[i + 1]:
      return arr[i]
  return arr[n - 1] | def check_element(list, element):
  flag = True
  for i in list:
    if (i != element):
      flag = False
      break
  return flag | def count(list1):
  c = 0
  for i in range(0,len(list1)):
    if list1[i] == True:
      c = c + 1
  return c |

Figure 7: Qualitative samples from the MBPP dataset retrieved through QUALEVAL.

Interestingly, the final example (right), generated by the davinci-2 model, generates a more robust solution than the ground truth! The ground truth solution assumes that the input is a list of booleans, while the model generation can accept any list with any data type. The test cases for this sample do not test for edge cases and therefore the ground truth program is technically correct.

## 5.3 MODEL IMPROVEMENT VIA QUALITATIVE EVALUATION: A CASE STUDY

We use the actionable insights from QUALEVAL to improve models on a variety of settings on the DialogSum dataset. We first leverage insights from QUALEVAL to precisely improve the proficiency on selected domains by fine-tuning a 13 billion parameter Llama 2 model (See A.1) Finally, we demonstrate QUALEVAL's potential for improving model performance through in-context learning.

**Model improvement via fine-tuning** For this setting, we try to simulate a real-world use case by choosing a relatively small training set. We first leverage QUALEVAL's flexible LP solver to also generate domain assignments for training samples. We choose a base set of 250 training samples. We then augment the training set by adding 250 samples from the "Employment/Professional skills", "Career/job interview" and "Social interactions" domains, with an equal proportion. For the baseline, we use the same training set but randomly augment the training set with 250 samples. We then train the off-the-shelf Llama model on both datasets and evaluate the proficiency of the model on the three selected domains. Figure 6 (right) visualizes the proficiency of the domains for both datasets and we find that the model trained on the augmented dataset is more proficient on all the augmented domains, with the ROUGE-2 score improving by $\approx 2$ percentage points across all domains. Therefore, the attribute assignments generated by QUALEVAL can be cleverly utilized to precisely improve model proficiency on selected attributes.

**Improved in-context learning** In-context learning is an important paradigm for customizing closed-source black box models. Therefore, selecting the appropriate incontext samples is crucial.

Moreover, practitioners rely on smaller models for faster and cheaper inference, and their often limited context windows further highlight the importance of selecting high-quality in-context samples.

We leverage QUALEVAL to generate high-quality in-context samples for the curie model. Due to the context window limitation of the curie model and the relatively long instances of DialogSum, we use a single few-shot in-context sample. We use our evaluator LLM, $\mathcal{E}$, to generate affinity scores of all domains for the evaluation samples, but we do not use the ground truth generations for this setting to avoid leaking the ground truth. We also calculate the domain affinity scores for all training samples and set the most aligned training sample for each evaluation sample as the in-context sample. Compared to randomly sampling in-context samples from the train set, our method improves ROUGE-2 by a percentage point ($10\%$ to $11\%$) and demonstrates QUALEVAL's versatility in improving models across a variety of settings – both fine-tuning and in-context.

## 6 RELATED WORK

**Model Debugging/Improvement** Prior work has attempted to address the problem of model debugging and improvement. Zhang et al. (2018) propose to evaluate different pairs of models on separate evaluation splits to understand model behavior. They also generate feature-level importance scores from "symptom" instances provided by humans. Graliński et al. (2019) introduce a model-agnostic method to find global features that "influence" the model evaluation score, allowing practitioners to exclude problematic features. Lertvittayakumjorn & Toni (2021) develop a framework to generate explanations for model predictions to allow humans to give feedback and debug models. Ribeiro et al. (2020) presents a framework to generate test cases at scale to evaluate model robustness, but constrains the test cases to be generated from simple templates and lexical transformations. Abid et al. (2022) propose a framework to generate counterfactual explanations for model errors to enable a better understanding of model behavior. Chen et al. (2023) introduce Self-Debugging, a method to enable a large language model to debug the predicted computer program through few-shot demonstrations. While these works provide limited insights into model behavior, they often require significant human intervention to understand model behavior and do not provide precise actionable insights for model improvement. Finally, these works are constrained to simple classification and regression tasks or single domains like code generation and do not provide a broad, task-agnostic, fully automated framework for model interpretation and improvement for real-world tasks.

**Automatic Evaluation of Machine Learning Models** Automatic evaluation metrics, based on lexical overlap, such as BLEU Papineni et al. (2002), ROUGE Lin (2004), METEOR Banerjee & Lavie (2005) have helped researchers evaluate and compare models on a variety of language tasks. Recent work has proposed to use machine learning models to evaluate other machine learning models. Methods like Zhang et al. (2019); Fu et al. (2023); Zhou et al. (2023) use pre-trained language models to evaluate the quality of generated text and therefore rely more or semantics than lexical overlap. While these automated metrics have expedited research progress by eliminating human effort from evaluation, they have limited evaluation to a single scalar metric and therefore fail to provide a holistic and comprehensive understanding of model performance.

**Issues with quantitative metrics** Multiple studies have pointed out that quantitative metrics are not sufficient to understand the behavior of LLMs and that they are not a good proxy for real-world performance Liu & Liu (2008); Novikova et al. (2017); Reiter & Belz (2009); Liu et al. (2016). While these studies advocate better quantitative metrics, the focus of our study is to propose a new framework based on *qualitative* evaluation.

## 7 CONCLUSION

We propose QUALEVAL, a qualitative evaluation framework that provides a comprehensive way of evaluating models with a keen eye on model improvement. Rather than rely on scalar quantitative metrics that ignore the nuanced behavior of the model, QUALEVAL thoroughly tests the model and provides actionable insights to improve the model iteratively. We show that these insights are indeed faithful and lead to up to 12% relative improvement. Our work is the first step towards building a data-scientist in a box for faster model iteration.

ETHICAL CONSIDERATIONS

Our work provides a potent way to ensure that certain tasks performed by data scientists can be automated. While this reduces the burden on them, it is also possible that it reduces the need to have a very large group of them on a certain project. This might have workforce implications. But the intention of the study is to show that with the current LLMs, we can improve evaluation by making it comprehensive.

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

## A  APPENDIX

### A.1  LLAMA FINE-TUNING

We use LoRA to efficiently fine-tune the Llama-13b model. We try two learning rates (2e-4 and 1e-3) and train for up to 300 steps with a batch size of 4.

### A.2  MISCELLANEOUS

We set N to be 15, p to be 4, and k to be 5 in our experiments.

### A.3  PROMPTS USED IN QUALEVAL

### A.4  DASHBOARDS

**MBPP**

Given the following examples, What are relevant domains for the following programs? Focus on the example programs BUT be general. Structure the response as a numbered list.

Given the example programs, What are specific ATOMIC sub-tasks a machine learning model need to be competent at for the underlying task? Focus on the example programs BUT be general. [IMPORTANT] Do NOT list the overall task as a subtask and be GENERAL. Structure the response as: Subtask:. Generate a numbered list.

**DialogSum**

Given the following conversations, What are relevant domains for the data? Focus on the example data BUT be general. Structure the response as a numbered list.

Given the example conversations, What are specific sub-tasks a machine learning model need to be competent at for the underlying task? Focus on the example data BUT be general. [IMPORTANT] Do NOT list the overall task as a subtask and be GENERAL. Structure the response as: Subtask:. Generate a numbered list.",

**MMLU(Clinical Knowledge)**

Given the following examples, What are relevant domains for the data? Focus on the example data BUT be general. Structure the response as a numbered list.

Given the example questions and answers on clinical biology, What are sub-tasks a machine learning model need to be competent at to be a good medical assistant. Focus on the example data BUT be please be general. For instance. [IMPORTANT] Do NOT list the overall task as a subtask and be GENERAL while being GROUNDED in the example data. Structure the response as: Subtask: <subtask>. Generate a numbered list.

Figure 8: Prompt used for discovering attributes

Given the input to a language model, Rate to what degree the input belong to each of the following domains. Rate on a scale of 1-5, with 5 being compeletely belongs and 1 being not belonging at all. \n. [Important] For each domain, format the output as, [Domain 1: <domain>, Score: <score>, Evidence: <Evidence for score>] 'n' [Domain 2: <domain>, Score: <score>, Evidence: <Evidence for score>] 'n' [Domain N: <domain>, Score: <score>, Evidence: <Evidence for score>]. [Important] Make sure to include concrete evidence based on the input to JUSTIFY the score. Remember you are an ACCURATE, FAITHFUL, CRITICAL and FAIR judge \n'.'",

Given the input to a language model, Rate to what degree each of the following subtasks are needed to successfully understand and complete the task. Rate on a scale of 1-5, with 5 being very used and 1 being not used at all. \n. [Important] For each subtask, format the output as [Subtask 1: <subtask>, Score: <score>, Evidence: <Evidence for score>] 'n' [Subtask 2: <subtask>, Score: <score>, Evidence: <Evidence for score>] 'n' [Subtask N: <subtask>,Score: <score>; Evidence: <Evidence for score>]. [IMPORTANT] Do NOT add '\n' between subtask, score and explanation. [Important] Make sure to include concrete evidence based on the input to JUSTIFY the score. Remember you are an ACCURATE, FAITHFUL, CRITICAL and FAIR judge \n'.

Figure 9: Prompt used for scoring attributes

**System**: Given a holistic picture of the performance of a machine learning model, you are asked to summarize the model's overall performance.

Given the above information, please write a brief summary highlighting important information. Please be precise and concise but please be comprehensive.

A machine learning model is tasked with the following task: {task_instruction}

These are the {subtasks/domains} for the task: {list of subtasks/domains}

In the evaluation data, these are the importance scores of the

Subtask/Domains: {json.dumps(prior probabilities of subtasks and domains)}

The following scores show how well the model performs on the subtasks/domains: {json.dumps(proficiency_scores of subtasks and domains)}

The following distance demonstrates how much the domains/subtasks are actually used for generating the output when they are requried to generate the input. Therefore, a low distance implies that the model is utilizing the category when it needs to: {json.dumps(correlation_scores of category)}. [Important] Lower distance implies the {category} is leveraged when it needs to be used.

Figure 10: Prompt used for generating insights

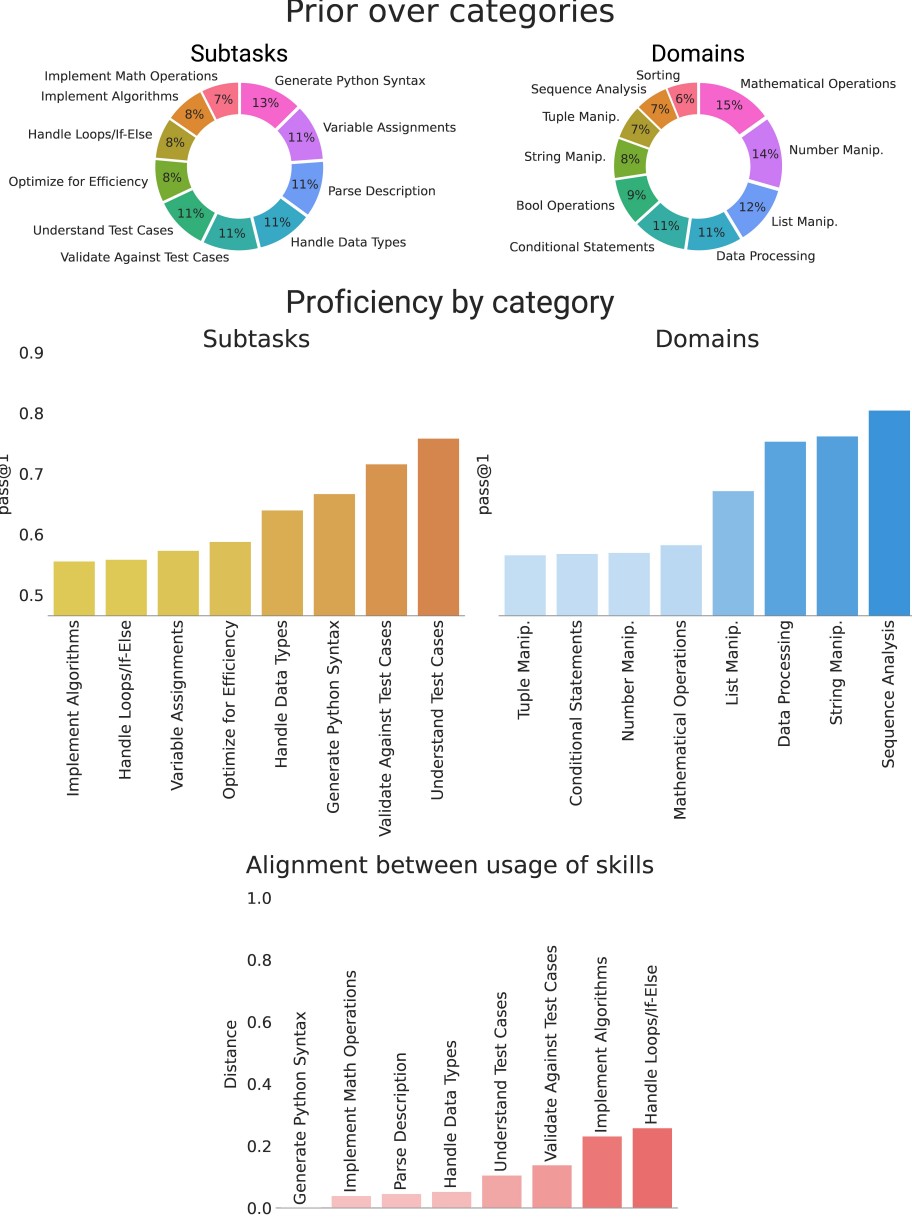

The machine learning model performs well on various subtasks, with the highest scores in "Understand test cases" and "Validate against test cases". It also excels in "Generate Python syntax" and "Manage variable assignments and data manipulation". However, it could improve in "Implement algorithmic operations" and "Handling loops and conditionals". The model effectively utilizes the subtasks when generating the output, particularly in "Generate Python syntax" and "Implement mathematical operations". In terms of domains, it performs strongly in "Sequence Analysis" and "String Manipulation", while improvements can be made in "Tuple Manipulation" and "Number Manipulation". Overall, the model demonstrates proficiency in understanding the requirements and generating accurate Python code, with potential for further enhancements in specific areas.

Figure 11: MBPP - davinci-2

# Prior over categories

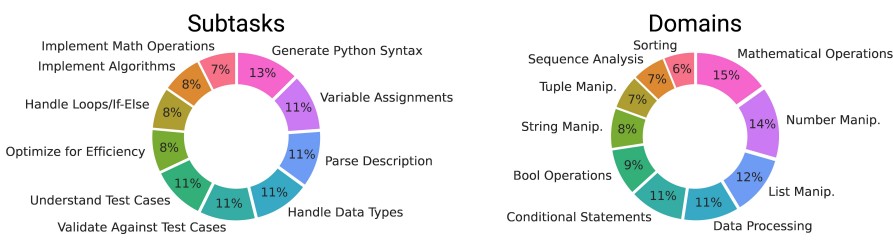

# Proficiency by category

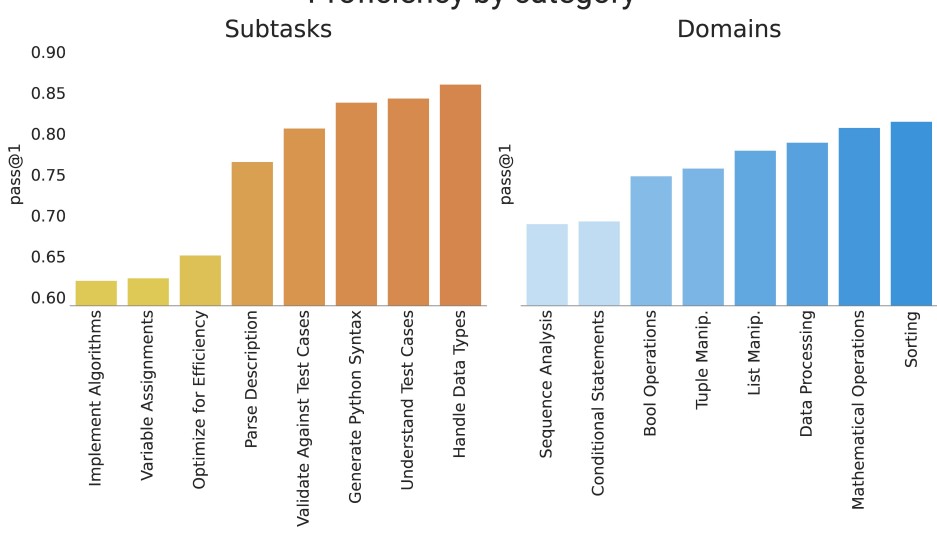

# Alignment between usage of skills

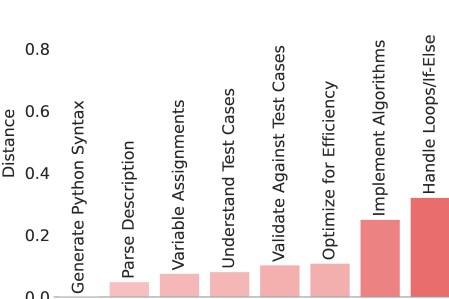

The machine learning model displays strong performance across various subtasks and domains. It accurately implements mathematical and algorithmic operations, handles loops and conditionals, and optimizes for efficiency and readability. The model demonstrates a good understanding of test cases and effectively validates against them. It effectively manages variable assignments and data manipulation and generates Python syntax with precision. Additionally, it shows a strong ability to parse natural language descriptions.

The model performs well in different domains, such as sorting, sequence analysis, tuple manipulation, string manipulation, boolean operations, conditional statements, data processing, list manipulation, number manipulation, and mathematical operations.

Furthermore, the model effectively utilizes subtasks when needed, as indicated by low distances between the required and actual usage of subtasks. This highlights its ability to leverage the necessary subtasks in generating the desired outputs.

Overall, the model exhibits a comprehensive understanding of the task and performs with high accuracy and efficiency, making it a reliable tool for generating Python functions from natural language descriptions and associated test cases.

Figure 12: MBPP - davinci-3

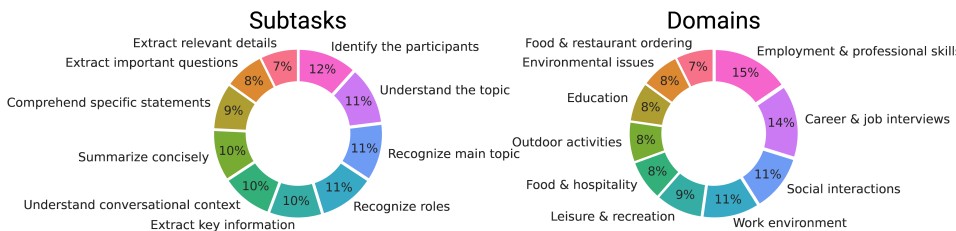

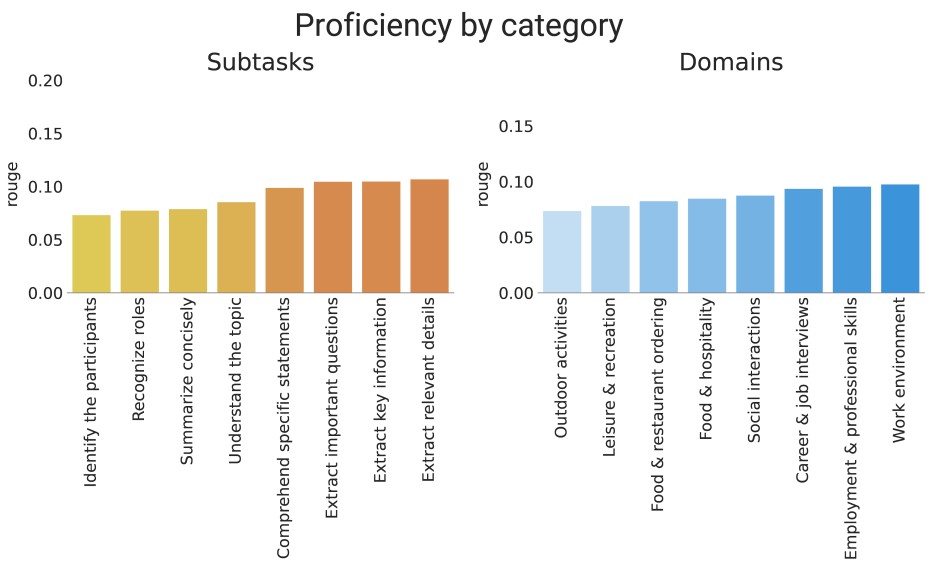

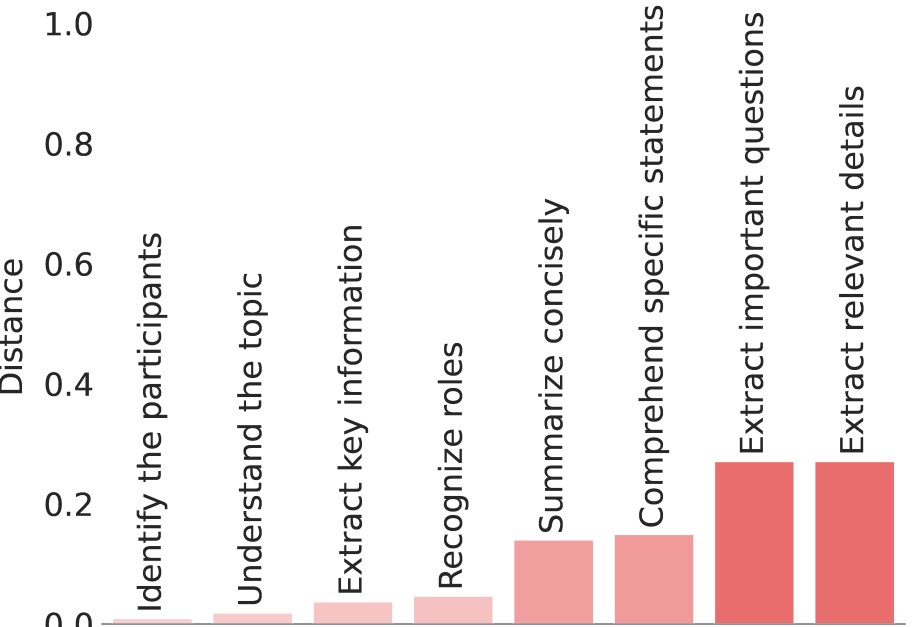

understanding the conversational context, recognizing the main topic, and comprehending specific statements. It also performs well in identifying participants and roles in the conversation. However, it struggles with summarizing conversations concisely and extracting relevant details.

In terms of domains, the model performs relatively well on social interactions, career interviews, and professional skills. However, it needs improvement in domains such as outdoor activities, leisure, food ordering, and environmental issues.

The model shows a significant gap between the importance assigned to certain subtasks and their actual utilization in generating the output. Notably, the extraction of relevant details and important information, as well as the understanding of specific statements and questions, are

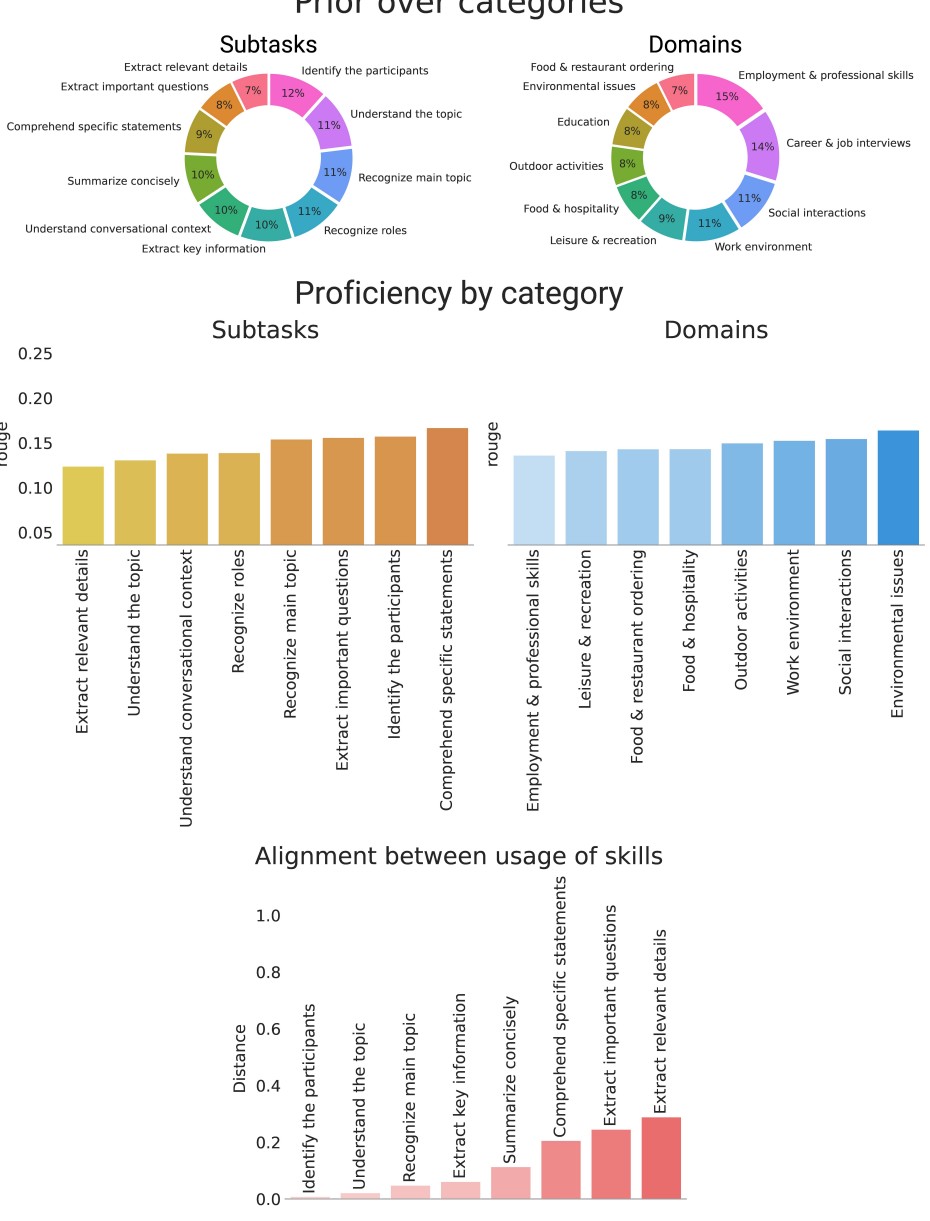

The machine learning model performs well on the overall task, with the highest scores for understanding the topic of discussion, recognizing the main topic of conversation, and identifying the participants in the conversation. It also effectively extracts key information and important details, comprehends specific statements or questions, and extracts relevant details. The model excels in understanding the conversational context and recognizing the roles and relationships of the speakers. It generates concise summaries and extracts important information and questions. In terms of domains, it performs particularly well in employment and professional skills, leisure and recreation, and food and restaurant ordering. However, there is room for improvement in environmental issues and pollution. The model demonstrates a strong utilization of the identified subtasks when required, with the highest utilization for identifying participants and understanding the topic of discussion.

Figure 14: DialogSum - davinci-2

# Prior over categories

## Subtasks

## Domains

# Proficiency by category

## Subtasks

## Domains

# Alignment between usage of skills

The machine learning model performs well in understanding the conversational context, recognizing the roles and relationships of speakers, and identifying the main topic of conversation. It also excels in extracting key information or important details and comprehending specific statements or questions. However, it struggles with summarizing the conversation concisely and extracting relevant details.

In terms of domains, the model performs well in leisure and recreation, food and hospitality, and environmental issues and pollution. It also shows good performance in career and job interviews, education, and work environment. However, it struggles in social interactions and personal relationships and food and restaurant ordering.

The model effectively utilizes the category of identifying the participants in the conversation and understanding the topic of discussion when generating the output. It also utilizes the category of extracting key information or important details and recognizing the main topic of conversation. However, it requires further improvement in leveraging the categories of summarizing the conversation concisely and extracting relevant details.

Figure 15: DialogSum - davinci-3

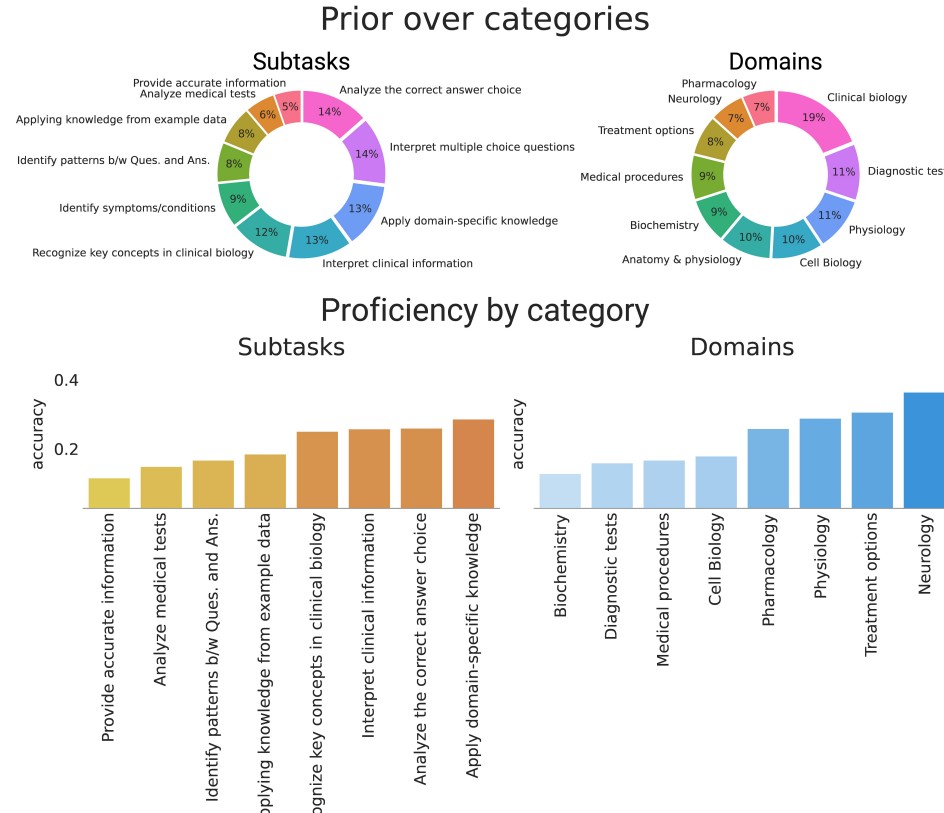

The machine learning model has been evaluated on various subtasks and domains related to clinical biology. Among the subtasks, the model performs relatively well in understanding and interpreting multiple choice questions, recognizing key terms and concepts in clinical biology, and applying domain-specific knowledge to select the most appropriate answer choice. However, it needs improvement in providing accurate and relevant information to healthcare professionals and patients, analyzing and processing medical test results, and identifying patterns and relationships between questions and answers.

In terms of domains, the model shows higher performance in physiology, treatment options, and pharmacology. On the other hand, it performs comparatively lower in biochemistry, diagnostic tests, and medical procedures and interventions. Notably, clinical biology has the highest importance score among the domains.

Overall, the model demonstrates a good understanding and interpretation of clinical information, but there is room for improvement in specific subtasks and domains to enhance its performance.

Figure 16: MMLU (Clinical Knowledge) - curie

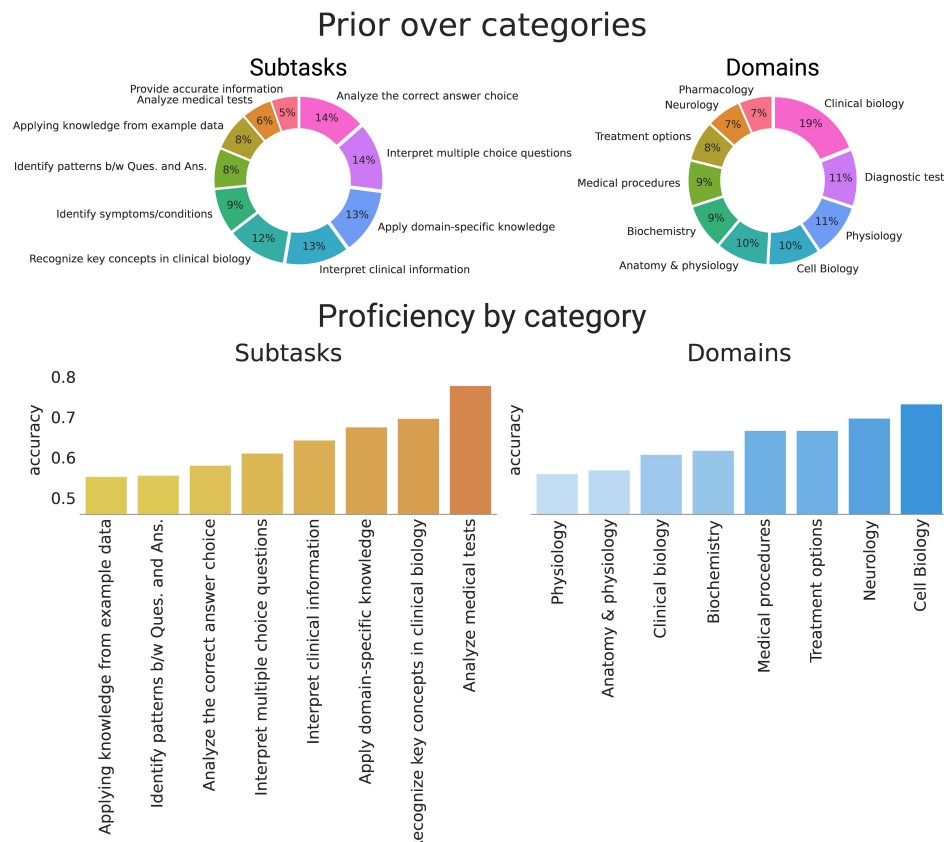

The machine learning model performed well on multiple subtasks, with the highest scores in analyzing and selecting the correct answer choice, understanding and interpreting multiple choice questions, and providing accurate and relevant information. It also excelled in applying domain-specific knowledge and recognizing key terms and concepts in clinical biology. The model showed good performance in retaining and applying knowledge from example data, identifying patterns and relationships between questions and answers, and interpreting clinical information. In terms of domains, the model performed strongly in clinical biology, diagnostic tests, pharmacology, and medical procedures and interventions. However, it had lower performance in neurology and physiology. Overall, the model demonstrated a solid understanding of clinical biology and was able to analyze and select the correct answer choice effectively.

Figure 17: MMLU (Clinical Knowledge) - davinci-2

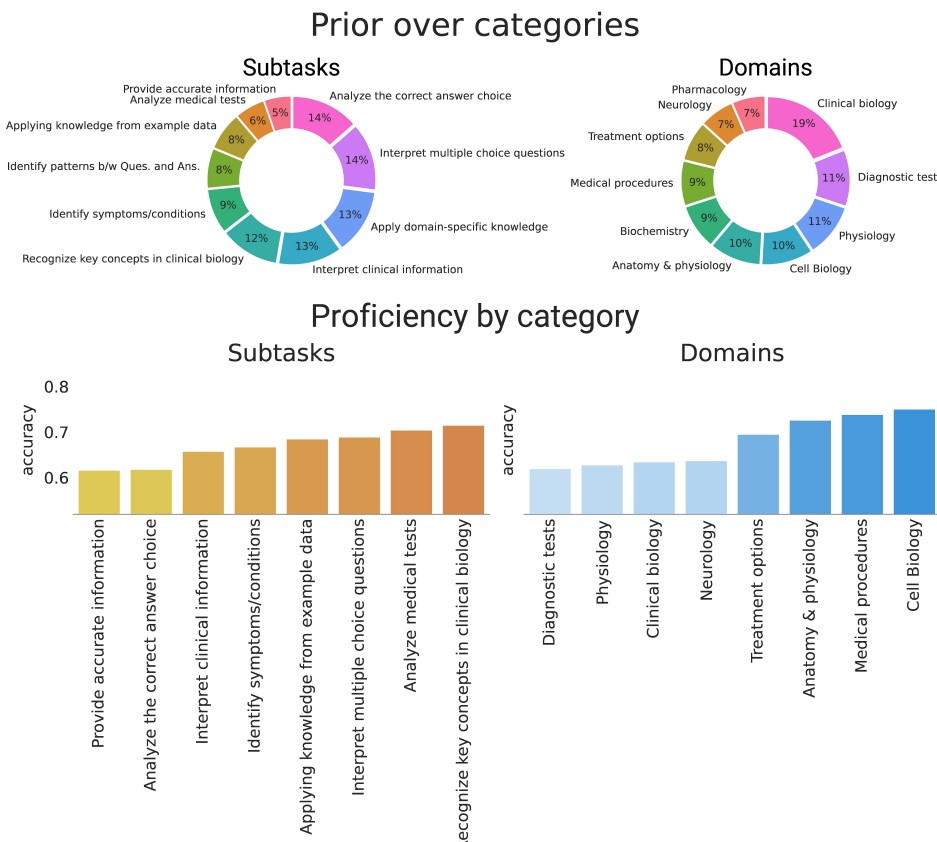

The machine learning model performs well across different subtasks and domains in clinical biology. It excels in understanding and interpreting multiple choice questions, analyzing and selecting the correct answer choice, and applying domain-specific knowledge. Recognizing key terms and concepts, as well as identifying patterns and relationships between questions and answers, are also strong areas for the model. It demonstrates good performance in understanding and interpreting clinical information and identifying symptoms, conditions, and diseases. The model performs best in analyzing and processing medical test results. Among the domains, it performs exceptionally well in cell biology, physiology, and medical procedures and interventions. The model also shows promising performance in anatomy and physiology, diagnostic tests, and treatment options.

Figure 18: MMLU (Clinical Knowledge) - davinci-3

