# OpenReview forum: "QualEval: Qualitative Evaluation for Model Improvement"
_ICLR.cc/2024/Conference — ICLR 2024 Conference Withdrawn Submission_

### Official Review · Reviewer_hZVP · 2023-10-31

**Soundness:** 3 good
**Presentation:** 2 fair
**Contribution:** 3 good
**Rating:** 3
**Confidence:** 2

**Summary:**

The paper introduces a technique/toolkit to enrich qualitative evaluations with assessments that use qualitative methods. The introduced QUALEVAL technique/toolkit provides a dashboard whose visualisations that help understand a given model and interrogate its performance. The QUALEVAL is evaluated using three models on three datasets. Quantitative evaluation metrics were included to supplement the evaluation.

**Strengths:**

The paper introduces a technique/toolkit to enrich qualitative evaluations with assessments that use qualitative methods. The introduced QUALEVAL technique/toolkit provides a dashboard whose visualisations that help understand a given model and interrogate its performance. The QUALEVAL is evaluated using three models on three datasets. Quantitative evaluation metrics were included to supplement the evaluation.

**Weaknesses:**

Although I am not an expert in introducing qualitative methods for evaluating models or algorithms, the paper seems to be somewhat lacking in scientific rigor in study design and reporting. I am not convinced that the contribution is complete what comes to its approach to providing evidence to validate the introduced technique/toolkit as a scholarly contribution. For example, what are the existing visualisation and dashboarding methods, how does the technique/toolkit go beyond them, and how is the experimental setup of the paper justifiable in connection with other similar studies?

Some more careful attention to detail is needed to perfect the paper. I found some typos (e.g., evalaute), missing punctuation marks (e.g., in equations), and incomplete/inconsistently formatted references (e.g., OpenAI. Introducing chatgpt, 2023. URL https://openai.com/blog/chatgpt). Also, text in some images is too small to read (e.g., Fig. 2 and 5).

**Questions:**

What are the existing visualisation and dashboarding methods?

How does the technique/toolkit go beyond them?

How is the experimental setup of the paper justifiable in connection with other similar studies?

---

### Official Review · Reviewer_BTXw · 2023-11-01

**Soundness:** 1 poor
**Presentation:** 2 fair
**Contribution:** 2 fair
**Rating:** 1
**Confidence:** 4

**Summary:**

This work proposes the idea of “quality over quantity” and designs a pipeline to analyze the quantitative results of LLMs and visualize the results with human-readable information. The empirical study shows the visualization and human-readable information can provide insights for developers to improve model performance further.

**Strengths:**

The research question of this work is important. If there is a way to go beyond quantitative results and provide insights of model performance, it will certainly reduce human effort on model analysis and improvement

**Weaknesses:**

The description of the pipeline is either ambiguous or problematic, and I have questions for basically every component described in section 2.2

- About attribute discovery
    - What is the definition of “domain” and “sub-task”, what is the difference between these two, and why these two are both called “attributes”?
    - While prompting LLM to list attributes, how do we know the results are reasonable and reliable? In other words, if we change the prompt a little bit, will we get different results? If the answer is yes, then how should we trust these results?
    - Given the procedure of iteratively getting more attributes, it sounds like there is a pre-defined number of attributes that we would like to get. Then, how to select this number? Does it depend any the specific dataset or task?
- About attribute assignment
    - Why two domains and sub-tasks for each instance? What do you mean by “concrete insights”? Are they the same “insights” that will later be generated by LLMs?
    - How to define the prior probability of an attribute?
    - I am not sure I understand this statement “To accommodate for the noisiness in an automated method, we make the prior probability constraint flexible by adding some slack …”
- About measuring sub-task skill alignment
    - Where we get the sub-tasks of the model prediction unless the model is working in a way that is similar to the chain of thoughts. However, this is unclear from the paper.
- About insight generation
    - I am seriously concerned about the reliability of generated “insights”. First of all, it is not clear to me what can be called an insight. Second, if this is similar to a summarization task, is hallucination an issue here, or is it not a problem at all?

**Questions:**

Please refer to the previous section

---

### Official Review · Reviewer_2Tzm · 2023-11-07

**Soundness:** 3 good
**Presentation:** 3 good
**Contribution:** 2 fair
**Rating:** 3
**Confidence:** 4

**Summary:**

The paper proposes a qualitative evaluation framework, called QUALEVAL, to better understand 1) different subtasks and domains in a training or a validation sets, and 2) how LLMs perform on them. Insights from these analysis can help improve 1) finetuning with additional data for tasks/domains where models were underperforming, or 2) few-shot prompts targeting examples from those domains. The paper demonstrates the efficacy of their framework on three datasets.


It is not clear whether or not the authors plan to release the code or the framework from this work.

**Strengths:**

QualEval clearly provides a more insightful analysis of the model performance on an eval set, rather than quantitative metrics such as Bleu or Rouge.


If the code/framework is made available, I can clearly see that the users will benefit from using this to automate the adaptation of LLMs for their usecases.

**Weaknesses:**

First of all, it is not clear whether or not the authors plan to release the code or the framework from this work.

As mentioned earlier, I believe that the framework will be valuable to analyze LLMs’ performance on a certain usecase and use the insights to better adapt them. However, I am afraid that the paper might not be a good fit for ICLR. Some of the findings that the model performance will improve with targeted in-context examples or finetuning data are not surprising and have been demonstrated before.

Also different eval sets annotated with ground truths are used to investigate LLMs performance. And then the same set is used to report the improvements which are not surprising as we are optimizing for these evalsets. I would be careful when referring to these improvements as model improvements, rather, we are adapting our models to do better on known use cases.

**Questions:**

Please see my comments in the Weakness section. Please address them if possible.

For the prompts in Figure 8 and 9, how much expert knowledge and prompt engineering are required? Is the framework flexible to allow such interventions?

---

### Comment · Reviewer_hZVP · 2023-11-20
**Summary**

Based on the received reviews and having not received author response, I cannot support accepting the paper. I do not see a reason to revise my original review either.